# Dietary Phytogenic Extracts Favorably Influence Productivity, Egg Quality, Blood Constituents, Antioxidant and Immunological Parameters of Laying Hens: A Meta-Analysis

**DOI:** 10.3390/ani12172278

**Published:** 2022-09-02

**Authors:** Arif Darmawan, Widya Hermana, Dwi Margi Suci, Rita Mutia, Anuraga Jayanegara, Ergin Ozturk

**Affiliations:** 1Department of Nutrition and Feed Technology, Faculty of Animal Science, IPB University, Bogor 16680, Indonesia; 2Department of Animal Science, Faculty of Agriculture, Ondokuz Mayis University, 55139 Samsun, Turkey; 3Animal Feed and Nutrition Modelling Research Group, Animal Science Faculty, IPB University, Bogor 16680, Indonesia

**Keywords:** bioactive compounds, egg, polyphenol, laying hens, meta-analysis

## Abstract

**Simple Summary:**

The application of phytogenic extracts in the poultry diet has been widely evaluated along with the prohibition on antibiotics use as growth promoters. Phytogenic extracts have been proven to improve the digestive health and performance of laying hens due to their antimicrobial, antioxidant, and immunomodulatory properties. However, several studies have also discovered negligible or even negative effects on the productive parameters or egg quality when added to diets at a high level. Through this meta-analysis approach, we found optimal levels of dietary phytogenic extracts that could be considered to avoid negative effects on laying hens. Furthermore, our findings support the suitability of phytogenic extracts for use as natural feed additives to increase the laying hens’ productivity with potential economic benefits.

**Abstract:**

The present study aimed to assess the impact of dietary phytogenic extracts on laying hen productivity, egg quality, blood constituents, antioxidant, and immunological parameters through a meta-analytical approach. A total of 28 articles (119 data points) reporting the influence of dietary phytogenic extracts on the productive performance, egg quality, blood constituents, immunological, and antioxidant parameters of laying hens were embedded into a database. Statistical analysis was performed using a mixed model, with different studies treated as random effects and phytogenic extract levels treated as fixed effects. This meta-analysis revealed that dietary phytogenic extracts quadratically (*p* < 0.05) improved egg production and egg mass as well as decreased (*p* < 0.05) the feed conversion ratio (FCR) with no adverse effect on egg weight and egg quality. Feed intake and egg yolk percentage tended to increase linearly (*p* < 0.1). Total serum cholesterol and low-density lipoprotein (LDL) declined quadratically (*p* < 0.001 and *p* < 0.05, respectively), high-density lipoprotein (HDL) increased linearly (*p* < 0.001), and malondialdehyde (MDA) decreased linearly (*p* < 0.01), with increasing levels of dietary phytogenic extract. In addition, immunoglobulin G (IgG), immunoglobulin A (IgA), glutathione peroxidase (GSH-Px), and total superoxide dismutase (TSOD) increased linearly (*p* < 0.05) in line with the increase in dietary phytogenic extract level. It was concluded that the inclusion of phytogenic extracts in the diet of laying hens had a positive effect on productive performance, feed efficiency, egg mass, immunity, and antioxidant activity without interfering with egg quality. The optimum level of feed photogenic extract for egg production and feed efficiency was determined to be around 300 mg/kg feed.

## 1. Introduction

The implementation of regulations prohibiting the application of antibiotics as growth promoters and growing concerns over the safety of poultry products have increased interest in the use of plant-based alternative feed additives. Phytogenic feed additives obtained from herbal plant extracts are commonly used in poultry, particularly in laying hens. Phytogenics, also known as phytobiotics, have beneficial effects on gut health and performance due to the presence of bioactive compounds such as polyphenols with antimicrobial, antioxidant, immunomodulatory, and anti-inflammatory properties [1,2]. Polyphenol compounds are the most widely produced plant bioactive compounds that serve to protect plants from the pests and UV radiation that can be found in plant parts, including the fruit, seeds, roots, bark, and leaves [3]. Flavonoids, phenolic acids, tannins, oligomeric proanthocyanidins, alkylresorcinols, avenanthramides, and lignans are some of the most well-known polyphenol groups [4].

Numerous studies on the diet of laying hens have confirmed the beneficial impact of phytogenic extracts on productive performance, egg quality, oxidative status, and immune system. However, several studies have also discovered a negligible effect of phytogenic extracts supplementation on the productive parameters or egg quality [5,6]. On the other hand, they may only produce a negative impact on poultry when added to diets at high levels [7,8]. For instance, a phenolic group such as tannins exhibits anti-nutritional properties at high concentrations. High levels of tannins (more than 10 g/kg feed) from plant extracts in the poultry diet can precipitate the protein and reduce fat digestion by binding bile salts or inactivating digestive enzymes [4,9,10]. Although several qualitative review articles have discussed differences in the responses of laying hens to dietary phytogenic extracts and a meta-analysis approach in broiler chickens [11,12], no meta-analysis has been performed to date in laying hens to quantify these differences. Therefore, the current meta-analysis study aimed to assess the impact of dietary phytogenic extracts on laying productivity, egg quality, blood constituents, and antioxidant and immunological parameters.

## 2. Materials and Methods

### 2.1. Database Development

Ethical approval was not required to conduct a meta-analysis study. The articles discussing the application of phytogenic extracts in laying hens were retrieved from scientific electronic databases such as Scopus, Web of Science, Crossref, Pub Med, Science Direct, and Google Scholar. To assist in the article selection process, search keywords such as “phytogenic”, “extract”, “phenolic”, “flavonoid”, and “laying hens” were applied. The following criteria were used to select articles for inclusion in the database: (a) performing in vivo trials on laying hens, (b) exclusively employing phytogenic extracts in the diets, (c) administration of extracts only through feed and without other confounding treatments, (d) reporting laying hens’ performance, egg quality, and blood parameters, and (e) the articles were written in English.

A total of 200 articles were initially found from the search engines based on the title and abstract of the article (Figure 1). The titles and abstracts of the articles were then screened based on the above-mentioned criteria, and 50 articles were eliminated for being improper, such as duplicate articles, review articles, or not being written in English. Finally, a total of 28 articles were added to the database for the meta-analysis after reviewing the substance including the data presentation, type of treatment, parameters observed, number of chickens, and proper statistical criteria.

### 2.2. Extraction and Description of Data

The information from the 28 selected articles is summarised in Table 1, including the authors’ names, publication year, strain, number and age of laying hens, extract level, plant name, plant part extracted, phytogenic content, and extract solvent type. Meanwhile, the variables included in the database were laying hen performances (egg production, feed intake, feed conversion ratio (FCR), egg weight, egg mass), egg qualities (eggshell thickness, eggshell strength, eggshell weight, egg yolk weight, albumen weight, Haugh unit, egg index, egg yolk colour, egg yolk cholesterol), and blood serum parameters (albumin, total protein, glucose, glycogen, aspartate aminotransferase (AST), total cholesterol, high-density lipoprotein (HDL), low-density lipoprotein (LDL), immunoglobulin G (IgG), immunoglobulin A (IgA), immunoglobulin M (IgM), total superoxide dismutase (TSOD), glutathione peroxidase (GSH-Px), and malondialdehyde (MDA)). Due to the limited number of studies, data on intestinal morphology and gut microbial population were not included in the database. Prior to the data processing, the parameter data were converted into similar units of measurement. The phytogenic extract level was reported in milligrams per kilogram of diet (mg/kg).

The articles (Table 1) were published between 2013 and 2022. A total of 7275 laying hens with a majority of Lohmann (28.6%) and Hy-line strain (25.0%) were used in the study. The plant materials used were leaf, bulb, seed, and peel that were extracted using water, ethanol, and petroleum ether as solvents. However, the types of solvent and plant material were not mentioned in several articles and/or were commercially used and were thus recorded as “unknown” in Table 1. The phytogenic extract levels ranged from 0 to 1000 mg/kg and were fed to laying hens aged 19 to 74 weeks.

### 2.3. Statistical Analysis

A mixed-model approach was used to analyze the data [13], which was performed in SAS^®^ On Demand for Academics using the MIXED PROC procedure. The different studies were treated as random effects and the phytogenic extract levels were treated as fixed effects. The statistical model was as follows:Y_ij_ = β_0_ + β_1_X_ij_ + β_2_X^2^_ij_ + *s*_i_ + *b*_i_ X_ij_ + *e*_ij_
where Y_ij_ = dependent variable, β_0_ = intercept in studies, β_1_ = coefficient of linear regression, β_2_ = coefficient of quadratic regression, X_ij_ = continuous variable predictor value (extract level), *b*_i_ = random effect of study *i* on the regression coefficient of Y on X, *s*_i_ = random effect of study, and *e*_ij_ = unexplained residual error.

Because the variable study contained no quantitative data, it was defined in the class expression. The corresponding linear regression model was used when the quadratic regression model was not significant. The *P*-value, the Akaike information criterion (AIC), and root mean square error (RMSE) were applied in the statistical model. The effect of treatment was considered significant at *p*-value < 0.05 and tended to be significant at *p* < 0.1.

**Table 1 animals-12-02278-t001:** Studies descriptions included in the database.

Author	Source	Main Bioactive Compound	Extract Level (mg/kg)	Chicken Breeds	Number of Birds	Age (Week)
Rahman et al. [8]	*Mentha piperita*	menthol, menthone, menthyl acetate	0–200	Babcock	252	21–30
Oh et al. [14]	*Diospyros kaki* L.	caffeic, p-coumaric, ferulic, gallic acids, tannins, terpenoids, naphthoquinones	0–750	Hy-lyne brown	120	50–56
Liu et al. [15]	commercial product	quercetin	0–600	Hessian	240	28–36
Ying et al. [16]	commercial product	quercetin	0–600	Hessian	240	29–38
Alagawany et al. [17]	*Yucca schidigera*	yuccaols, resveratrol	0–150	Hi-sex-brown	96	36–52
Ahmed et al. [18]	*Olea europaea* L.	hydroxytyrosol, vanillin, rutin, caffeic acid, catechin	0–150	Bandarah	360	24–42
Iskender et al. [19]	commercial product	hesperidin, naringin, quercetin	0–500	Lohmann white	96	29–40
Damaziak et al. [20]	*Allium sativum* L., *Allium cepa* L.	alicin, quercitin, gallic acid	0–32	ISA Brown	216	22–32
Sun et al. [21]	grape seed	procyanidins, proanthocyanidins	0–150	Hy-Line brown	640	25–33
Vakili and Heravi [22]	*Thymus vulgaris* L., *Foeniculum vulgare*	thymol, carvacrol (*Thymus vulgaris*); anethole, limonene, fenchone, estragole, safrole, camphene (*Foeniculum vulgare*)	0–40	Hy-Line	200	26–38
Park et al. [23]	*Trigonella foenum-graecum* L.	4-hydroxy isoleucine, trigonelline, carotenoids, coumarins, saponins	0–1000	Hy-Line brown	96	36–52
Simitzis et al. [24]	commercial product	quercetin	0–700	Lohmann brown-classic	192	70–74
Damaziak et al. [25]	*Zingiber officinale*, *Thymus vulgaris*	gingerol, sholaol (*Zingiber officinale*); borneol, thymol, carvacrol (*Thymus vulgaris*)	0–32	ISA brown	216	19–35
Xie et al. [26]	*Lonicera confusa*, *Astragali radix*	luteolin, chlorogenic acid, caffeic acid (*Lonicera confusa*); astragaloside, formononetin, calycosin (*Astragali radix*)	0–1000	Lohmann pink-shell	1440	52–64
Song et al. [27]	*Camelia oleifera*	glucuronic acid, xylose, rhamnose, methyl pentose	0–500	Hy-Line brown	180	26–38
Huang et al. [28]	*Camellia sinensis* (L.) *O. Ktze.*	theanine, theobromine, caffeine, catechins	0–300	Lohmann brown	240	30–38
Huang et al. [29]	*Rhizoma drynariae*	naringin, neoeriocitrin, triterpenes, phenylpropanoids	0–200	Lohmann pink-shell	216	54–67
Dos Santos et al. [30]	*Psidium cattleianum* Sabine	ellagic acid, gallic acid, catechin, quercetin	0–200	ISA Brown	75	45–49
Kılınç and Karaoğlu [31]	*Hypericum perforatum* L.	hypericin, hyperforin, flavonoids	0–300	Lohmann white	336	40–52
Liu et al. [32]	*Curcuma longa*	curcumin	0–200	Hy-Line brown	240	40–46
Mutlu and Yildirim [33]	*Panax ginseng*	saponin glycosides (ginsenosides), essential oils sterols, flavonoids	0–150	Atak-S brown	80	28–32
Widjastuti et al. [34]	*Garcinia mangostana* L.	xanthone, flavonoids, anthocyanins	0–240	Sentul	40	20–32
Wen et al. [35]	*Zingiber officinale* Roscoe	6-gingerol, 8-gingerol, 10-gingerol	0–100	Hyline Brown	288	40–48
Abad et al. [36]	*Allium* spp	alicin, quercitin, gallic acid	0–700	Lohmann Brown	180	36–40
Zhu et al. [37]	Neohesperidin	neohesperidin	0–400	Lohmann	240	66–74
Guo et al. [38]	*Macleaya cordata*	sanguinarine, chelerythrine	0–200	Xuefeng black-bone	576	47–59
Peng et al. [39]	*Eucommia ulmoides*	chlorogenic acid, aucubin, geniposidic acid	0–500	Spotted-brown	120	56–67
Guo et al. [40]	*Pinusmassoniana* Lamb	flavonoids, shikimic acid	0–400	Peking pink	60	50–58

## 3. Results

### 3.1. Productive Performances and Egg Quality

Dietary phytogenic extract quadratically increased (*p* < 0.05) egg production (Figure 2) and egg mass, and it quadratically (*p* < 0.05) decreased FCR. Feed intake tended to increase linearly (*p* < 0.1); however, the inclusion of phytogenic extracts did not affect the egg weight (Table 2). Based on the egg production and FCR parameters, the optimum phytogenic extract levels for laying hens were 292 mg/kg and 313 mg/kg feed, respectively. In general, the administration of phytogenic extracts did not affect the egg qualities (eggshell weight, eggshell thickness, eggshell strength, egg yolk colour, egg index, albumen weight, Haugh unit). However, the egg yolk weight percentage tended to increase linearly (*p* < 0.1) (Table 3).

### 3.2. Blood Constituents and Egg Yolk Cholesterol

The effect of phytogenic extracts on blood constituents and egg yolk cholesterol concentration is presented in Table 4. Serum cholesterol and LDL concentrations declined quadratically (*p* < 0.001 and *p* < 0.05, respectively) with increasing dietary phytogenic extract levels, whereas HDL concentration increased linearly (*p* < 0.05). Meanwhile, egg yolk cholesterol concentration tended to decrease linearly (*p* < 0.1). On the other hand, phytogenic extracts supplementation did not affect total protein, glucose, albumin, ALT, and AST.

### 3.3. Immunological and Antioxidant Parameters

The relationship between phytogenic extract level and immunological and antioxidant parameters is presented in Table 5. The IgM concentration was not affected by the addition of the phytogenic extract. However, the IgG, IgA, TSOD, and GSH-Px concentrations increased linearly (*p* < 0.05) with an increase in the dietary phytogenic extract level. Similarly, the concentration of MDA decreased linearly (*p* < 0.01) with increasing levels of phytogenic extracts.

## 4. Discussion

Phytogenics, also known as phytobiotics, are plant bioactive compounds that have beneficial effects on gastrointestinal health and the performance of poultry due to the presence of phytogenic compounds such as polyphenols with antioxidant and immunomodulatory properties [1,41]. Various studies have been conducted to assess the efficacy of phytogenic feed additives in laying hens to minimise antibiotic use. Beneficial effects on productive performance and egg quality were obtained by supplementing the laying hen’s diet with *Pinus massoniana* [40], *Curcuma longa* [32], *Geranium thunbergia*, and *Mentha arvensi* [42]. However, egg production, FCR, and egg weight were not increased with the addition of dietary *Thymus vulgaris* L. [43] and *Mentha piperita* [8]. Sharma et al. [6] stated that the administration of garlic and thyme to the diets of laying hens did not increase egg weight.

Our meta-analysis study generally revealed that dietary phytogenic extracts showed a positive effect on productivity, blood metabolites, and immunological and antioxidant parameters with no adverse effect on egg quality. Phytogenic compounds improved poultry performance by increasing digestive enzyme secretion, lowering the number of pathogenic bacteria in the digestive tract, or modulating intestinal morphology functions [4]. Previously, Iqbal et al. [44] and Tellez-Isaias et al. [45] confirmed that polyphenols can suppress several bacterial pathogens, including *Salmonella enteritidis* and *E. coli*. Similar results have shown that quercetin inclusion, one of the flavonoid compounds, enhanced the productive performance of laying hens due to its ability to reduce intestinal pathogenic bacteria [5,46]. This claim was supported by Mutlu et al. [47] who stated that the inclusion of quercetin can reduce coliforms in the cecal of laying quail and increase the *lactobacilli* population. In addition, the inclusion of curcumin has been reported to induce antibacterial activity through the inhibition of bacterial cell proliferation by interfering with the GTPase of the FtsZ protofilament activity, which was critically involved in bacterial cell division and survival [48]. In the case of gut health, Abdel-Moneim et al. [5] and Prihambodo et al. [11] revealed that flavonoids in herbal plants have a favorable effect on the digestive tract of poultry. They argued that flavonoids have antioxidant properties and can enhance the function of the small intestine in nutritional absorption. Other phytogenic compounds, such as genistein and hesperidin, also had a beneficial effect on gut morphology, including villus density, crypt depth, and villus height [49]. Then, higher villi increase the surface area of the intestine and improve nutrient absorption, whereas deeper crypts promote rapid villi renewal in response to pathogen-induced inflammation [50]. However, the limitations of this meta-analysis have not been able to confirm gut health due to the limited number of studies related to gut morphology and gut microbial populations of laying hens.

Based on the present meta-analysis, TSOD, GSH-Px, IgG, and IgA increased linearly in line with the increasing levels of dietary phytogenic extract. Under oxidative stress, the poultry body is unable to efficiently eliminate excess free radicals, particularly reactive nitrogen species and reactive oxygen species. Enzymatic mechanisms, including triad catalase, GSH-Px, and TSOD, are one mechanism for the removal of these free radicals [51,52]. Meanwhile, polyphenols are external antioxidants that serve as the first defense for cells against excessive free radical production and protect their constituents from oxidative damage. Among all polyphenols, flavonoids are the most effective at eliminating free radicals and preventing their negative effects [53,54]. For instance, naringenin and naringin have strong scavenging activity for lipid peroxidation inhibitors [55]. Furthermore, rutin, hesperidin, and genistein supplementation were found to improve GSH-Px, SOD, and T-AOC activity and decrease MDA serum concentrations [56,57]. Thus, these findings suggest that rutin, genistein, and hesperidin have the capacity to stimulate antioxidant enzymes, reduce oxidative stress, and further reduce MDA concentration in the blood. Generally, the mechanisms of polyphenol in protecting the cells from free radical oxidation include the activation of antioxidant enzymes (e.g., SOD, GPH-Px), pro-oxidant enzymes inhibition such as xanthine oxidase, direct cleaning of ROS by donating electrons, and an increase in the antioxidant activity of antioxidant substances (e.g., ascorbate, tocopherol) [58]. Phytogenic extracts also provide promising immunotherapy due to the considerable increase in IgG and IgA concentrations. Recent studies have found that polyphenol modulates immune cell activity by binding to cellular receptors, modulating cell signalling pathways, and thus controlling host immunological responses [59]. For instance, tea polyphenols and curcumin raised the total antibody-secreting cells in the spleen and significantly improved immunoglobulin levels and humoral immune response [60,61]. Meanwhile, according to Abd El Latif et al. [62], the increase in immunoglobulin value following the addition of herbal plant supplementation rich in flavonoids prolonged the activity of other antioxidant properties such as vitamin C.

This meta-analysis revealed that supplementing the diets of laying hens with phytogenic extracts lowered serum cholesterol, LDL, and improved HDL. Flavonoids can also reduce low-density lipoprotein cholesterol peroxidation by minimizing plasma and membrane lipid oxidation [63]. Zhou et al. [64] found that using flavonoid baicalein feed additive for broilers can reduce serum cholesterol and LDL. Wang et al. [60] reported that tea polyphenols reduced TC and LDL levels in serum due to increased cholesterol excretion through the excreta. The liver produces endogenous cholesterol and is transferred to extrahepatic tissues by LDL. Meanwhile, HDL transports cholesterol from peripheral tissues to the liver before excreting it via the bile pathway [65]. In addition, polyphenols induce the expression of the cholesterol enzyme 7-alpha hydroxylase, which controls the bile synthesis and homeostasis of cholesterol and inhibits the activity of hydroxyl-3-methyl-glutaryl-CoA as a limiting enzyme for cholesterol synthesis [66,67]. Moreover, egg cholesterol deposition is closely related to plasma triglyceride level, total cholesterol, and LDL [68]. Our meta-analysis approach confirmed the presence of lower levels of TC and LDL in plasma, as well as a linear tendency to decrease egg yolk cholesterol levels with higher dietary levels of phytogenic extracts.

Generally, while polyphenolic compounds have a beneficial impact at a certain level, several studies have reported a negative effect on the performance of poultry when polyphenols were added to poultry diets at high levels [7]. This decline in poultry performance may be attributed to the decreased digestion of fats and proteins through the binding of bile salts and/or inactivation of digestive enzymes. Meanwhile, the presence of polyphenol substances such as condensed tannins, which bind bile salts and restrict fat digestion, and the ability of polyphenols to bind endogenous proteins to form insoluble complexes may be related to the inhibition of digestive enzymes [5,9]. Therefore, the optimal level of dietary phytogenic extract identified through this meta-analysis approach can be considered to avoid these negative effects on laying hens.

## 5. Conclusions

The current meta-analysis confirms that the inclusion of phytogenic extracts in laying hens aged 19–74 weeks has a positive effect on productive performance, feed efficiency, and egg mass without interfering with egg quality. The optimum level of dietary phytogenic extract for egg production and feed efficiency is around 300 mg/kg diet. The phytogenic extracts have beneficial effects as antioxidant and immunomodulating agents demonstrated by an increase in TSOD, GSH-Px, IgA, IgG, and a decrease in oxidation products (MDA) in serum.

## Figures and Tables

**Figure 1 animals-12-02278-f001:**
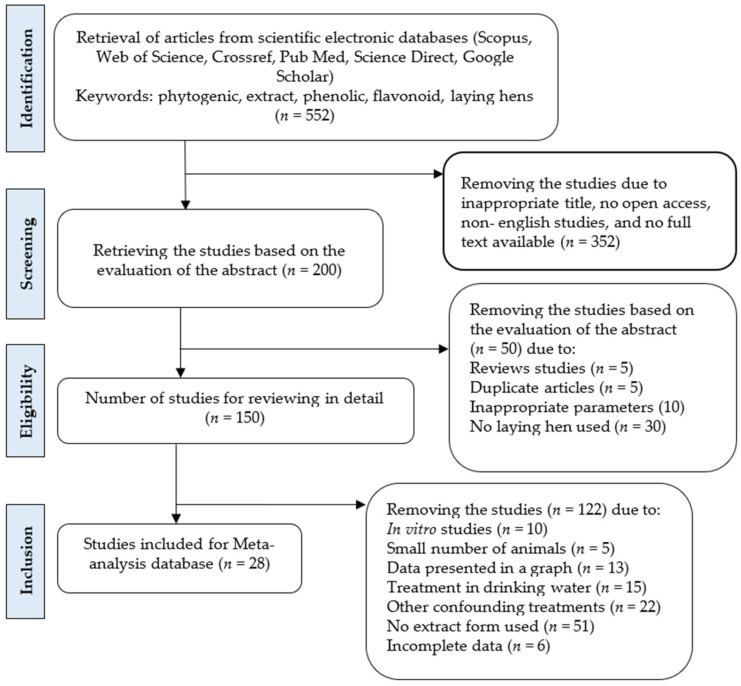
Flowchart of identification, screening, and inclusion process of meta-analysis database.

**Figure 2 animals-12-02278-f002:**
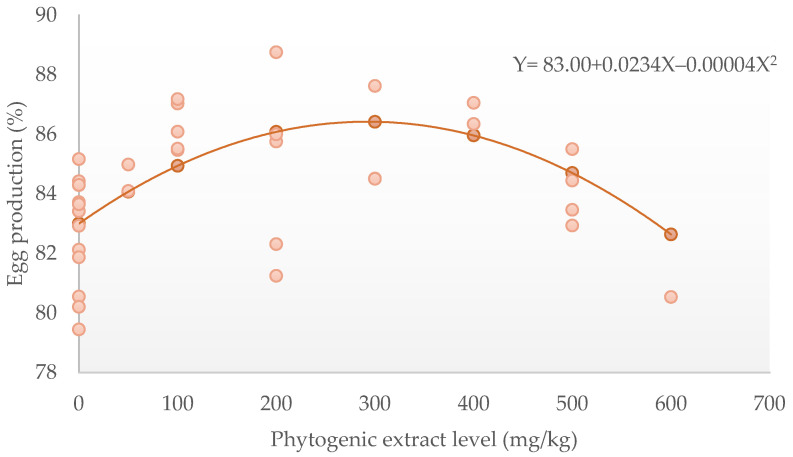
Quadratic equation of dietary phytogenic extract (mg/kg) to egg production (%).

**Table 2 animals-12-02278-t002:** Regression equations for the impact of phytogenic extract levels on productive performances of laying hens.

Parameter	*n*	Intercept	SE Intercept	Slope	SE Slope	*p*-Value	RMSE	AIC	Model	Trend
Egg production (%)	72	83	1.77	0.0234	0.007055					
				−0.00004	0.000014	0.02	7.25	457	Q	Positive
Feed intake (g/hen/day)	97	112	2.32	0.00315	0.00177	0.08	6.16	576	L	Positive
FCR	94	2.1	0.049	−0.00027	0.000185					
				0.000000431	0.00000012	<0.001	0.21	−36	Q	Negative
Egg weight (g/egg)	102	61.2	0.78	0.000948	0.000639	0.14	2.21	349	L	-
Egg mass (g/hen/day)	101	49.6	2.47	0.0119	0.00388					
				−0.00002	0.0000074	0.03	4.38	450	Q	Positive

*n*, treatment number; RMSE, root mean square error; AIC, Akaike information criterion; SE, standard error; Q, quadratic; L, linear; FCR, feed conversion ratio.

**Table 3 animals-12-02278-t003:** Regression equations for the impact of phytogenic extract levels on the egg quality of laying hens.

Parameter	*n*	Intercept	SE Intercept	Slope	SE Slope	*p*-Value	RMSE	AIC	Model	Trend
Eggshell thickness (mm)	99	0.36	0.0056	0.000011	0.00000777	0.16	0.04	422	L	-
Eggshell strength (Newton)	92	37.4	0.91	0.00119	0.00118	0.32	5.86	514	L	-
Albumen weight (%)	21	60.8	1.88	−0.00022	0.00155	0.89	2.41	93.3	L	-
Egg yolk weight (%)	42	27.2	1.21	0.000672	0.000367	0.08	1.31	155	L	Positive
Eggshell weight (%)	33	12.7	0.66	0.00102	0.001104	0.37	1.32	102	L	-
Haugh unit	119	85	1.41	0.00167	0.00157	0.29	7.34	559	L	-

*n*, treatment number; RMSE, root mean square error; AIC, Akaike information criterion; SE, standard error; L, linear.

**Table 4 animals-12-02278-t004:** Regression equations for the impact of phytogenic extract levels on egg yolk cholesterol and blood parameters of laying hens.

Parameter	*n*	Intercept	SE Intercept	Slope	SE Slope	*p*-Value	RMSE	AIC	Model	Trend
Egg yolk cholesterol (mg/g)	20	13.4	0.9	−0.0132	0.00676	0.08	4.04	118	L	Negative
Serum cholesterol (mg/dL)	54	151	7.73	−0.168	0.0318					
				0.000239	0.000048	<0.001	44.6	488	Q	Negative
LDL (mg/dL)	36	50.7	8.49	−0.0473	0.0128					
				0.000042	0.000015	0.01	19.2	304	Q	Negative
HDL (mg/dL)	37	34	7.38	0.00657	0.0028	0.03	12.6	284	L	Positive
Total protein (g/L)	42	54	2.44	−0.00396	0.00515	0.45	11.9	288	L	-
Glucose (mg/dL)	25	204	22	−0.0195	0.0302	0.53	22.2	216	L	-
Albumin (g/dL)	21	2.33	0.13	−0.00028	0.000328	0.4	0.31	9	L	-
AST (U/L)	26	205	21.1	−0.0292	0.0279	0.31	41.1	246	L	-
ALT (U/L)	23	2.64	0.69	−0.00028	0.00103	0.8	0.97	66.1	L	-

*n*, treatment number; RMSE, root mean square error; AIC, Akaike information criterion; SE, standard error; Q, quadratic; L, linear; HDL, high-density lipoprotein; LDL, low-density lipoprotein; AST, aspartate aminotransferase; ALT, alanine aminotransferase.

**Table 5 animals-12-02278-t005:** Regression equations for the impact of phytogenic extract levels on immunological and antioxidant parameters of laying hens.

Parameter	*n*	Intercept	SE Intercept	Slope	SE Slope	*p*-Value	RMSE	AIC	Model	Trend
IgG (mg/dL)	22	3.56	0.75	0.00176	0.000593	0.01	0.74	38.7	L	Positive
IgM (mg/dL)	20	33.2	10.92	0.01073	0.01103	0.36	13.6	106.4	Q	-
IgA (mg/dL)	21	38.6	15.9	0.0158	0.00388	0.002	4.84	94.6	L	Positive
TSOD (U/mL)	33	194	18.8	0.0491	0.018	0.01	32.7	310	L	Positive
GSH-Px (U/mL)	28	7.56	0.86	0.0029	0.00122	0.03	7.4	160	L	Positive
MDA (nmol/mL)	21	4.21	0.16	−0.00093	0.00024	0.002	1.44	59.3	L	Negative

*n*, treatment number; RMSE, root mean square error; AIC, Akaike information criterion; SE, standard error; Q, quadratic; L, linear; IgG, immunoglobulin G; IgA, immunoglobulin A; IgM, immunoglobulin M; TSOD, total superoxide dismutase; GSH-Px, glutathione peroxidase; MDA, malondialdehyde.

## Data Availability

The data presented in this study are available on request from the corresponding author.

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
