# Peer review of "Dietary Phytogenic Extracts Favorably Influence Productivity, Egg Quality, Blood Constituents, Antioxidant and Immunological Parameters of Laying Hens: A Meta-Analysis"

_animals, 2022, doi:10.3390/ani12172278_

Round 1
Reviewer 1 Report
Darmawan and others assessed the impact of dietary phytogenic extracts on laying hen productivity, egg quality, blood constituents, antioxidant, and immunological parameters through a meta-analytical approach. Based on this meta-analysis authors found a favorable influence on productivity, egg quality, blood constituents, antioxidant and immunological parameters. The manuscript is well written and I recommend publication of this nice work. I have some „Minor comments“. Line 40: bioactive compound should be bioactive compounds; Lines 63-65: Please mention the dose of tannins that might negatively impact poultry. Figure 1: studiies …should be studies. Line 104: please expand the full name of FCR. Line 142: This Productive …should be Productive (please delete the word „this„?). Line 212: Salmonella …should be italic. Line 214: Please cite this recent review „ https://doi.org/10.51585/gjvr.2021.3.0014“.
Author Response
Dear
Reviewers of Animals
Sincerest thanks for giving us the opportunity to submit a revised of our manuscript titled “Dietary Phytogenic Extracts Favorably Influence Productivity, Egg Quality, Blood Constituents, Antioxidant and Immunological Parameters of Laying Hens: A Meta-Analysis (Animals 1863470) to Animals. We appreciate the time and effort that you have dedicated to providing your valuable feedback on our manuscript. We are grateful to the reviewers for their insightful comments on our paper. We have been able to incorporate changes to reflect most of the suggestions provided by the reviewers. We have highlighted the changes within the manuscript.
Here is a point-by-point response to the reviewers’ comments and suggestions.
Best regards
Arif Darmawan
Response to Reviewer 1 Comments
Point 1: Darmawan and others assessed the impact of dietary phytogenic extracts on laying hen productivity, egg quality, blood constituents, antioxidant, and immunological parameters through a meta-analytical approach. Based on this meta-analysis authors found a favorable influence on productivity, egg quality, blood constituents, antioxidant and immunological parameters. The manuscript is well written and I recommend publication of this nice work. I have some „Minor comments“. Line 40: bioactive compound should be bioactive compounds; Lines 63-65: Please mention the dose of tannins that might negatively impact poultry. Figure 1: studiies …should be studies. Line 104: please expand the full name of FCR. Line 142: This Productive …should be Productive (please delete the word „this„?). Line 212: Salmonella …should be italic. Line 214: Please cite this recent review „ https://doi.org/10.51585/gjvr.2021.3.0014“.
Response 1: Thank you very much for your positive responses and suggestion to our article. We have corrected all notes in our manuscript
- Line 40: we have amended bioactive compound to be bioactive compounds (line 45)
- Lines 63-65: we have mentioned the level of tannin (line 68)
- Figure 1: studiies : we have amended studiies to be studies
- Line 104: please expand the full name of FCR : we have amended FCR to be feed conversion ratio (FCR) (line 110)
- Line 212: Salmonella …should be italic : we have amended Salmonella Enteritidis to be Salmonella enteritidis (line 220)
- Line 214: Please cite this recent review „ https://doi.org/10.51585/gjvr.2021.3.0014“. : we have cited the article. (line 219)
Response to Reviewer 2 Comments
The application of phytogenic extracts in the poultry diet has been increasingly used with the ban on the use of antibiotics as growth promoters. These compounds have been shown to improve the digestive health and performance of laying hens, but some studies have also shown negative effects on production parameters or egg quality. The submitted review article aimed to evaluate the impact of phytogenic extracts from the diet on laying hen productivity, egg quality, blood constituents, antioxidants and immunological parameters through a meta-analytic approach. This type of study is necessary to obtain new data on this nutritional practice in poultry farms. Therefore, the issue of review is very important to demonstrate alternatives to the use of antimicrobial products.
In my opinion, the manuscript review is also well written and an important contribution in the field of poultry nutrition. However, I have some concerns as pointed out below:
1. Meta-analysis reviews must be rigorous in the search procedure. The authors presented the flowchart that describes the different stages of this search. However, the search criteria were not defined exactly. In addition, why were not included some previous articles cited in the Discussion (such as references 41, 42, 44, etc.)? Therefore, I suggest that authors describe the selection process more accurately (what combinations of words were used in all databases). This is very important to demonstrate the absence of any selection bias in this procedure. It is also necessary for anyone else to replicate the same I think it is not necessary to demonstrate all the criteria in the main body of the manuscript. I suggest including as supplementary material.
Response 1:
Thank you very much for comments and suggestions. We have improved the flowchart to be more accurate, including the number of included and excluded articles and the selection process. The reason for removing the article has been explained in the criteria and flowchart. For example, we excluded reference 41 because plant extracts are administered through drinking water (number of excluded articles = 15 as "Treatment in drinking water" in the flowchart) and it does not include our criteria whereby extracts should be administered through feed (criteria b, line 86). Also, in reference 42, the treatments are applied in powder form (number of excluded articles = 51 as "No extract form used" in flowchart) and do not include our criteria whereby we only included phytogenic extracts form (criteria c, line 86). Meanwhile, reference 44 is not an open-access journal making us unable to access all data (already mentioned in the flow chart).
For the criteria, we have summarized them clearly in a flowchart so we think it is already clear even though we don't add it in the supplementary material.
2. The flowchart should also have more exact information and numbers. For example, it is very subjective to just describe "Evaluating the title and abstract". More precise information is necessary. Please review all flowchart to demonstrate the searching procedure as well as the number of the articles included in the study in more detail.
Response 2
Thank you very much for the suggestion. We have revised the flowchart to be more accurate, covering the number of included and excluded articles as well as the selection process.
3. Table 1 is too large and it was not properly prepared. I suggest to remove columns “No” (with the numbers of the articles from 1 to 28), “material” and “solvent”. If necessary, the readers can evaluate the material and solvent in the original articles. In addition, it is necessary to describe the name of the plant source instead of “commercial product” (included in many lines) as well as to substitute “chicken strains” by “chicken breeds”.
Response 3
Thank you very much for the suggestion. We agree with your suggestion and have amended Table 1 according to the suggestions. However, we can not describe the name of the plant source in Table 1, because the cited references only mention the commercial product without mentioning the plant source.
4. Results in Tables can be presented in Figures, such as bar graphs. Differences could be more easily noticed.
Response 4
Thank you very much for the suggestion. In this meta-analysis, We emphasize the effectiveness of phytogenic extracts on each parameter by looking at the relationship between phytogenics and parameters (linear or quadratic regression) so that we can estimate the optimal phytogenic dose. We evaluated the P value to see the significance of the phytogenic inclusion relationship with the parameters. To choose the best model, we used the RMSE and AIC values. The lower the RMSE and AIC, the better the model. The effect of treatment is considered significant at p-value < 0.05 and tended to be significant at p < 0.1 (mentioned in lines 135 to 136). For clarity, we have added a graph of the quadratic equation for egg production (figure 2).
5. The Discussion has to be additionally reviewed after all modifications. I also suggest the inclusion of a paragraph with the limitations of this meta-analysis study.
Response 5
Thank you very much for the suggestion. We have stated the limitations of this meta-analysis on lines 231 to 233
6. I would recommend to prepare a shorter text with more concise and direct Conclusions.
Response 6
Thank you very much for the suggestion. We have amended the conclusion to be shorter with more concise (lines 289to 294)
7. After all modifications, the Abstract need to be revised too.
Therefore, the manuscript needs a lot more work before it is ready for another peer review.
Response 7
Thank you very much for the suggestion. We have revised the abstract and added P value

Reviewer 2 Report
The application of phytogenic extracts in the poultry diet has been increasingly used with the ban on the use of antibiotics as growth promoters. These compounds have been shown to improve the digestive health and performance of laying hens, but some studies have also shown negative effects on production parameters or egg quality. The submitted review article aimed to evaluate the impact of phytogenic extracts from the diet on laying hen productivity, egg quality, blood constituents, antioxidants and immunological parameters through a meta-analytic approach. This type of study is necessary to obtain new data on this nutritional practice in poultry farms. Therefore, the issue of review is very important to demonstrate alternatives to the use of antimicrobial products.
In my opinion, the manuscript review is also well written and an important contribution in the field of poultry nutrition. However, I have some concerns as pointed out below:
1) Meta-analysis reviews must be rigorous in the search procedure. The authors presented the flowchart that describes the different stages of this search. However, the search criteria were not defined exactly. In addition, why were not included some previous articles cited in the Discussion (such as references 41, 42, 44, etc.)? Therefore, I suggest that authors describe the selection process more accurately (what combinations of words were used in all databases). This is very important to demonstrate the absence of any selection bias in this procedure. It is also necessary for anyone else to replicate the same procedure. I think it is not necessary to demonstrate all the criteria in the main body of the manuscript. I suggest including as supplementary material.
2) The flowchart should also have more exact information and numbers. For example, it is very subjective to just describe "Evaluating the title and abstract". More precise information is necessary. Please review all flowchart to demonstrate the searching procedure as well as the number of the articles included in the study in more detail.
3) Table 1 is too large and it was not properly prepared. I suggest to remove columns “No” (with the numbers of the articles from 1 to 28), “material” and “solvent”. If necessary, the readers can evaluate the material and solvent in the original articles. In addition, it is necessary to describe the name of the plant source instead of “commercial product” (included in many lines) as well as to substitute “chicken strains” by “chicken breeds”.
4) Results in Tables can be presented in Figures, such as bar graphs. Differences could be more easily noticed.
5) The Discussion has to be additionally reviewed after all modifications. I also suggest the inclusion of a paragraph with the limitations of this meta-analysis study.
6) I would recommend to prepare a shorter text with more concise and direct Conclusions.
7) After all modifications, the Abstract need to be revised too.
Therefore, the manuscript needs a lot more work before it is ready for another peer review.
Author Response
Dear
Reviewers of Animals
Sincerest thanks for giving us the opportunity to submit a revised of our manuscript titled “Dietary Phytogenic Extracts Favorably Influence Productivity, Egg Quality, Blood Constituents, Antioxidant and Immunological Parameters of Laying Hens: A Meta-Analysis (Animals 1863470) to Animals. We appreciate the time and effort that you have dedicated to providing your valuable feedback on our manuscript. We are grateful to the reviewers for their insightful comments on our paper. We have been able to incorporate changes to reflect most of the suggestions provided by the reviewers. We have highlighted the changes within the manuscript.
Here is a point-by-point response to the reviewers’ comments and suggestions.
Best regards
Arif Darmawan
Response to Reviewer 1 Comments
Point 1: Darmawan and others assessed the impact of dietary phytogenic extracts on laying hen productivity, egg quality, blood constituents, antioxidant, and immunological parameters through a meta-analytical approach. Based on this meta-analysis authors found a favorable influence on productivity, egg quality, blood constituents, antioxidant and immunological parameters. The manuscript is well written and I recommend publication of this nice work. I have some „Minor comments“. Line 40: bioactive compound should be bioactive compounds; Lines 63-65: Please mention the dose of tannins that might negatively impact poultry. Figure 1: studiies …should be studies. Line 104: please expand the full name of FCR. Line 142: This Productive …should be Productive (please delete the word „this„?). Line 212: Salmonella …should be italic. Line 214: Please cite this recent review „ https://doi.org/10.51585/gjvr.2021.3.0014“.
Response 1: Thank you very much for your positive responses and suggestion to our article. We have corrected all notes in our manuscript
- Line 40: we have amended bioactive compound to be bioactive compounds (line 45)
- Lines 63-65: we have mentioned the level of tannin (line 68)
- Figure 1: studiies : we have amended studiies to be studies
- Line 104: please expand the full name of FCR : we have amended FCR to be feed conversion ratio (FCR) (line 110)
- Line 212: Salmonella …should be italic : we have amended Salmonella Enteritidis to be Salmonella enteritidis (line 220)
- Line 214: Please cite this recent review „ https://doi.org/10.51585/gjvr.2021.3.0014“. : we have cited the article. (line 219)
Response to Reviewer 2 Comments
The application of phytogenic extracts in the poultry diet has been increasingly used with the ban on the use of antibiotics as growth promoters. These compounds have been shown to improve the digestive health and performance of laying hens, but some studies have also shown negative effects on production parameters or egg quality. The submitted review article aimed to evaluate the impact of phytogenic extracts from the diet on laying hen productivity, egg quality, blood constituents, antioxidants and immunological parameters through a meta-analytic approach. This type of study is necessary to obtain new data on this nutritional practice in poultry farms. Therefore, the issue of review is very important to demonstrate alternatives to the use of antimicrobial products.
In my opinion, the manuscript review is also well written and an important contribution in the field of poultry nutrition. However, I have some concerns as pointed out below:
1. Meta-analysis reviews must be rigorous in the search procedure. The authors presented the flowchart that describes the different stages of this search. However, the search criteria were not defined exactly. In addition, why were not included some previous articles cited in the Discussion (such as references 41, 42, 44, etc.)? Therefore, I suggest that authors describe the selection process more accurately (what combinations of words were used in all databases). This is very important to demonstrate the absence of any selection bias in this procedure. It is also necessary for anyone else to replicate the same I think it is not necessary to demonstrate all the criteria in the main body of the manuscript. I suggest including as supplementary material.
Response 1:
Thank you very much for comments and suggestions. We have improved the flowchart to be more accurate, including the number of included and excluded articles and the selection process. The reason for removing the article has been explained in the criteria and flowchart. For example, we excluded reference 41 because plant extracts are administered through drinking water (number of excluded articles = 15 as "Treatment in drinking water" in the flowchart) and it does not include our criteria whereby extracts should be administered through feed (criteria b, line 86). Also, in reference 42, the treatments are applied in powder form (number of excluded articles = 51 as "No extract form used" in flowchart) and do not include our criteria whereby we only included phytogenic extracts form (criteria c, line 86). Meanwhile, reference 44 is not an open-access journal making us unable to access all data (already mentioned in the flow chart).
For the criteria, we have summarized them clearly in a flowchart so we think it is already clear even though we don't add it in the supplementary material.
2. The flowchart should also have more exact information and numbers. For example, it is very subjective to just describe "Evaluating the title and abstract". More precise information is necessary. Please review all flowchart to demonstrate the searching procedure as well as the number of the articles included in the study in more detail.
Response 2
Thank you very much for the suggestion. We have revised the flowchart to be more accurate, covering the number of included and excluded articles as well as the selection process.
3. Table 1 is too large and it was not properly prepared. I suggest to remove columns “No” (with the numbers of the articles from 1 to 28), “material” and “solvent”. If necessary, the readers can evaluate the material and solvent in the original articles. In addition, it is necessary to describe the name of the plant source instead of “commercial product” (included in many lines) as well as to substitute “chicken strains” by “chicken breeds”.
Respose 3
Thank you very much for the suggestion. We agree with your suggestion and have amended Table 1 according to the suggestions. However, we can not describe the name of the plant source in Table 1, because the cited references only mention the commercial product without mentioning the plant source.
4. Results in Tables can be presented in Figures, such as bar graphs. Differences could be more easily noticed.
Response 4
Thank you very much for the suggestion. In this meta-analysis, We emphasize the effectiveness of phytogenic extracts on each parameter by looking at the relationship between phytogenics and parameters (linear or quadratic regression) so that we can estimate the optimal phytogenic dose. We evaluated the P value to see the significance of the phytogenic inclusion relationship with the parameters. To choose the best model, we used the RMSE and AIC values. The lower the RMSE and AIC, the better the model. The effect of treatment is considered significant at p-value < 0.05 and tended to be significant at p < 0.1 (mentioned in lines 135 to 136). For clarity, we have added a graph of the quadratic equation for egg production (figure 2).
5. The Discussion has to be additionally reviewed after all modifications. I also suggest the inclusion of a paragraph with the limitations of this meta-analysis study.
Response 5
Thank you very much for the suggestion. We have stated the limitations of this meta-analysis on lines 231 to 233
6. I would recommend to prepare a shorter text with more concise and direct Conclusions.
Response 6
Thank you very much for the suggestion. We have amended the conclusion to be shorter with more concise (lines 289to 294)
7. After all modifications, the Abstract need to be revised too.
Therefore, the manuscript needs a lot more work before it is ready for another peer review.
Response 7
Thank you very much for the suggestion. We have revised the abstract and added P value

Round 2
Reviewer 2 Report
The manuscript can be accepted for publication.